# Bioinformatics of Differentially Expressed Genes in Phorbol 12-Myristate 13-Acetate-Induced Megakaryocytic Differentiation of K562 Cells by Microarray Analysis

**DOI:** 10.3390/ijms23084221

**Published:** 2022-04-11

**Authors:** Seung-Hoon Lee, Na Rae Park, Jung-Eun Kim

**Affiliations:** 1Department of Molecular Medicine, School of Medicine, Kyungpook National University, Daegu 41944, Korea; jsat1234@naver.com (S.-H.L.); nrpark85@naver.com (N.R.P.); 2BK21 Four KNU Convergence Educational Program of Biomedical Sciences for Creative Future Talents, Department of Biomedical Science, Kyungpook National University, Daegu 41944, Korea; 3Cell and Matrix Research Institute, Kyungpook National University, Daegu 41944, Korea

**Keywords:** megakaryocytes, microarray data, differentially expressed genes, STRING

## Abstract

Megakaryocytes are large hematopoietic cells present in the bone marrow cavity, comprising less than 0.1% of all bone marrow cells. Despite their small number, megakaryocytes play important roles in blood coagulation, inflammatory responses, and platelet production. However, little is known about changes in gene expression during megakaryocyte maturation. Here we identified the genes whose expression was changed during K562 leukemia cell differentiation into megakaryocytes using an Affymetrix GeneChip microarray to determine the multifunctionality of megakaryocytes. K562 cells were differentiated into mature megakaryocytes by treatment for 7 days with phorbol 12-myristate 13-acetate, and a microarray was performed using RNA obtained from both types of cells. The expression of 44,629 genes was compared between K562 cells and mature megakaryocytes, and 954 differentially expressed genes (DEGs) were selected based on a *p*-value < 0.05 and a fold change >2. The DEGs was further functionally classified using five major megakaryocyte function-associated clusters—inflammatory response, angiogenesis, cell migration, extracellular matrix, and secretion. Furthermore, interaction analysis based on the STRING database was used to generate interactions between the proteins translated from the DEGs. This study provides information on the bioinformatics of the DEGs in mature megakaryocytes after K562 cell differentiation.

## 1. Introduction

Megakaryocytes are large hematopoietic cells present in the bone marrow cavity and are primarily responsible for platelet production [1]. Although the proportion of megakaryocytes among the total number of bone marrow cells is very small, they are involved in several essential functions, including blood coagulation and inflammatory responses [2,3]. Megakaryocytes are distributed in the yolk sac, liver, and spleen during development [4,5] and have functions in the lungs [6]. As megakaryocytes secrete various cytokines and growth factors, they are expected to have a wide range of effects on themselves and in the surrounding cellular environment [7,8].

Megakaryocytes are immune cells that participate directly in inflammation and immunity [2]. During viral infection, megakaryocytes promote antiviral immunity through interferon-induced transmembrane protein 3, thus inhibiting viral invasion [9]. Furthermore, lung megakaryocytes function as immune cells by presenting antigens to CD4+ T cells in response to bacterial infection [10]. Megakaryocytes are also involved in separating blood and lymphatic vasculatures through platelets during embryonic development [11]. Although the proportion of megakaryocytes in the body is quite low, they play several roles by harmoniously interacting with the transcription of numerous genes and the expression of proteins throughout the body. However, studies seeking to identify the genes or proteins that are differentially expressed during megakaryocyte generation, differentiation, and maturation are lacking. K562 human leukemia cells are widely used as a model of hematopoietic differentiation and are differentiated into hematopoietic lineage cells by various differentiation-inducing agents [12,13,14]. Among these agents, phorbol 12-myristate 13-acetate (PMA) was most commonly used to differentiate K562 cells into megakaryocyte lineages [15,16]. Therefore, we identified the overall changes in gene expression patterns during K562 differentiation into megakaryocytes by PMA using microarray analysis.

Differential gene expression analysis is important for understanding the biological differences between cells and condition dependence. Microarray analysis is a common method for differential gene expression analysis [17]. Nowadays, RNA-Seq, a more sensitive and accurate tool than microarrays, is frequently used to detect differentially expressed genes (DEGs), but its limitations, including high costs, high data storage requirements, and a time-intensive process, need to be overcome before it can be established as the predominantly used method for transcriptome analysis [18]. Therefore, microarrays are still used in clinical diagnostic testing for diseases as they are cost-effective for large-scale studies [19]. Moreover, microarrays can be useful considering their highly standardized protocols for target preparation, hybridization, processing, and excellent reproducibility [20]. This study screened numerous genes from mature megakaryocytes through microarray analysis, and genes showing distinct changes were classified into gene ontology categories. Consequently, this analysis is expected to explain the multifunctionality of megakaryocytes.

## 2. Results

### 2.1. Differentiation of K562 Cells into Megakaryocytes

K562 leukemia cells were differentiated into megakaryocytes using PMA treatment for 5 or 7 days, which is widely used for megakaryocyte commitment and differentiation [12,13]. DMSO, a known inducing agent for megakaryocyte differentiation, was used for the control treatment [14]. We performed quantitative real-time PCR (qPCR) and cytological examinations to confirm differentiation into mature megakaryocytes. Cell morphology was compared between K562 cells before and after differentiation using PMA treatment with cytological staining. At day 7, PMA-induced megakaryocytes were found to be increased in size and have multilobed nuclei (Figure 1A). The diameter distribution of PMA-induced megakaryocytes was in the range of 6 to 50 μm (average 18.23 μm), whereas the distribution of K562 and DMSO-treated cell diameters was in the range of 4 to 20 μm (average 10.13 μm) and 6 to 20 μm (average 11.92 μm), respectively (Figure 1B). All K562 and DMSO-treated cells were less than 20 µm in diameter, whereas more than 30% of PMA-induced megakaryocytes were greater than 20 µm in diameter (Figure 1B). On day 5 of differentiation, only 4.7% of PMA-induced megakaryocytes were greater than 20 µm in diameter. Most of the cell diameter distribution was similar between the three groups (Appendix A). On day 7 of differentiation, PMA-induced megakaryocytes were more adherent to the culture plate and were more spread out than K562 or DMSO-treated cells (Figure 1C). Increased adhesion of PMA-induced megakaryocytes was also observed on differentiation day 5 compared to K562 or DMSO-treated cells (Appendix A). Additionally, PMA-induced megakaryocytes at day 7 showed not only 2N ploidy but also 4N to 16N ploidy, whereas K562 cells had only 2N and 4N ploidy and DMSO-treated cells had 2N to 8N ploidy, indicating an increased polyploidization level of PMA-induced megakaryocytes (Figure 1D). However, only 2N and 4N ploidy were observed in all types of cells on day 5 of differentiation (Appendix A). When K562 cells differentiate into megakaryocytes for 7 days, the megakaryocyte marker genes GATA1, GATA2, and *NF-E2*, are upregulated [21]. In contrast, the lymphoid-primed progenitor markers, CD34 and *c-KIT*, are downregulated [22]. Compared with K562 cells, GATA1 and GATA2 expressions were significantly increased, whereas CD34 and *c-KIT* expressions were decreased in PMA-induced megakaryocytes (Figure 1E). However, the expressions of CD41 (also known as *ITGA2B*) and *GP1B*, other markers for late-stage megakaryocytes [23], were not upregulated in PMA-induced megakaryocytes compared with the K562 cells (Figure 1E). Interestingly, although similar expression patterns were observed in GATA1, *c-KIT*, CD41, and *GP1B*, GATA2 and CD34 expressions were not altered in DMSO-treated cells compared to PMA-induced megakaryocytes (Figure 1E). The expression of CD235A, which is expressed in erythroid precursors and erythrocytes, was decreased in PMA-induced megakaryocytes and DMSO-treated cells compared with K562 cells (Appendix A). Finally, these results indicate that K562 cells were differentiated into megakaryocytes as a result of PMA treatment for 7 days.

### 2.2. Functional Enrichment Analysis of DEGs in Mature Megakaryocytes

We performed a microarray analysis to compare the DEGs between K562 cells and PMA-induced megakaryocytes. The expressions of 44,629 genes between K562 cells before and after differentiation into megakaryocytes were compared (Appendix A). Among them, 954 genes were either upregulated or downregulated by >2 fold. These DEGs were classified into 12 distinct functional categories according to gene ontology: cell differentiation, immune response, secretion, neurogenesis, cell migration, inflammatory response, cell death, extracellular matrix, apoptotic signaling, cell cycle, angiogenesis, and aging (Figure 2A). The pie chart represents the percentages of genes in each functional category (Figure 2A). When differentiating into megakaryocytes, genes related to cell differentiation (22%) and immune response (16%) were highly affected, whereas those related to secretion, neurogenesis, cell migration, and inflammatory response were overrepresented by 7–11%. Genes in the other categories were overrepresented by <5% and affected to a lesser extent by differentiation into megakaryocytes. Five functional categories were selected for further analysis (Figure 2B). In the inflammatory response category, 42 genes were significantly differentially expressed, of which 9 were upregulated and 33 were downregulated. In the angiogenesis category, 22 genes were significantly differentially expressed; 4 were upregulated and 18 were downregulated. The cell migration category included 49 DEGs of 13 upregulated and 36 downregulated genes. There were 30 DEGs in the extracellular matrix category; 12 were upregulated and 18 were downregulated. The secretion category included 66 DEGs, of which 28 were upregulated and 38 were downregulated.

### 2.3. DEG Profiles and Protein–Protein Network Construction in Mature Megakaryocytes

The expressions of genes included in the five selected categories were displayed as a heat map with normalized z-scores. An illustration of the interaction between proteins translated from genes displayed in each heat map was generated by interaction analysis based on the STRING database. Several proteins that did not interact with the central cluster of each category were excluded. The protein–protein network consisted of nodes, colored as red to blue representing upregulation to downregulation. The thickness of the interconnections between proteins represented the level of evidence for each interaction.

Inflammatory response: Megakaryocytes play a direct role in inflammation and innate immunity [2]. In particular, megakaryocytes assist in the primary function of lymphocytes for antigen presentation by antigen-presenting cells [24]. Therefore, it is crucial to analyze the differential expression of genes related to the inflammatory response during differentiation into megakaryocytes. Among the nine upregulated genes, the expressions of *IL5RA*, *FN1*, and *IL20RB* were mainly increased. On the other hand, among the 33 downregulated genes, the expressions of *CCR4*, *IL1A*, *CCL2*, and CD44 were primarily decreased (Figure 3A). The protein–protein network for inflammatory response-related genes consisted of 33 nodes, indicating that proteins increased or decreased by K562 cell differentiation into megakaryocytes (Figure 3B). Among genes that were differentially expressed, the expressions of upregulated genes *IL5RA*, *IL20RB*, and *LYZ* and downregulated genes *CCR4, IL1A,* and *CCL2* were verified by qPCR (Figure 3C).

Angiogenesis: Megakaryocytes provide angiogenic factors that promote the secretion of other factors, such as TGF-β1 [25,26]. The expressions of *FN1*, *UNC5B*, *VEGFA*, and *LEF1* were increased, whereas those of *ITGB3*, *CCL2*, and *HEY1* were primarily decreased among the 18 downregulated genes (Figure 4A). The protein–protein network of angiogenesis-related genes consisted of 18 nodes, indicating proteins increased or decreased by K562 cell differentiation into megakaryocytes (Figure 4B). To validate microarray analysis data, among genes showing significant expression differences, we selected five genes that are related to angiogenesis and carried out qPCR. *FN1* and *VEGFA* expressions were upregulated, and *ITGB3*, *HEY1*, and *GPR4* expressions were downregulated in megakaryocytes (Figure 4C).

Cell migration: Megakaryocytes reach the terminal stage of differentiation to form platelets [27]. After performing various functions in the bone marrow cavity, they move to bone marrow blood vessels to form platelets. This movement requires cell migration based on motility [28]. Therefore, the differential expression of genes related to cell migration may provide clues as to the biological function of mature megakaryocytes. Among the 13 upregulated genes, *FN1*, *MAP1B*, and *BVES* were mainly increased, whereas, among the 36 downregulated genes, *ITGB3*, *CCR4*, *LRP12*, and *CCL2* were primarily decreased (Figure 5A). The protein–protein network in the cell migration category consisted of 40 nodes, indicating proteins increased or decreased by K562 cell differentiation into megakaryocytes (Figure 5B). The expressions of four selected genes which are related to cell migration were verified by qPCR. *MAP1B* expression was upregulated, and *LRP12*, CD44, and *ITGA9* expressions were downregulated in megakaryocytes (Figure 5C).

Extracellular matrix: The extracellular matrix supports tissues and regulates intercellular communication, and it is essential for cellular processes that include angiogenesis and cell migration [29]. In addition, megakaryocytes express extracellular matrix components, including fibronectin, collagen, and laminin [30]. It will be helpful to analyze the differential expression of extracellular matrix-related genes to explain the relationship among the extracellular matrix, angiogenesis, and cell migration during K562 cell differentiation into megakaryocytes. The expressions of *SERPINF1*, *FN1*, and *COL6A5* were remarkably increased among the 12 upregulated genes, whereas those of *CILP*, *TFPI2*, and *A2M* were decreased among the 18 downregulated genes (Figure 6A). The protein–protein network of the cell migration-related genes consisted of 20 nodes, indicating that proteins increased or decreased by K562 cell differentiation into megakaryocytes (Figure 6B). Among genes that were differentially expressed, the expressions of the upregulated genes *SERPINF1*, *COL6A5*, and *PRG2* and the downregulated genes *CILP*, *TFPI2*, and *A2M* were verified by qPCR (Figure 6C).

Secretion: Megakaryocytes play a role in various tissues, either directly or indirectly, through numerous secreted factors and signaling pathways [31]. Therefore, the differential expression of genes related to secretion was analyzed between K562 cells and megakaryocytes. The expressions of *ANXA3*, *FN1*, and *TCN1* were remarkably increased among the 28 upregulated genes, whereas those of *ITGB3*, *SUCNR1*, *CGA*, *SCG3*, *RAB27B*, and *STOM* were primarily decreased among 38 downregulated genes (Figure 7A). The protein–protein network in the secretion-related cluster consisted of 16 and 24 nodes, indicating that proteins increased and decreased, respectively, by K562 cell differentiation into megakaryocytes (Figure 7B). The expressions of four selected genes which are related to secretion were verified by qPCR. *TCN1*, *SLC18A2*, and *S100P* expressions were upregulated, and *CGA* expression was downregulated in megakaryocytes (Figure 7C).

## 3. Discussion

Megakaryocytes are cells found in the bone marrow and are responsible for several essential functions that maintain homeostasis in the body, including platelet production, blood coagulation, and inflammatory responses [2,3,32]. However, to understand the multiple functions of megakaryocytes in the body, it is necessary to study genes or proteins that are differentially expressed in the generation, differentiation, and maturation of megakaryocytes. Using microarray analysis, we screened for global changes in gene expression during the differentiation of K562 human leukemia cells into megakaryocytes, classified the DEGs into gene ontology categories, and analyzed protein–protein networks based on the STRING database. Through this study, we provide information regarding the multiple influences that megakaryocytes exert on the surrounding cellular environment. The changes in the cell cycle, death, and differentiation during megakaryocyte differentiation are autonomous processes. Additionally, since megakaryocytes play the role of immune cells, it is natural that the expressions of genes related to the immune response change during differentiation into megakaryocytes. Excluding these categories, we finally selected five functional categories—inflammatory response, angiogenesis, cell migration, extracellular matrix, and secretion—to study the multiple functions of megakaryocytes. Principal component analysis (PCA) was also used to analyze microarray data and derive information from data simplified through the t-SNE algorithm. Additional studies with PCA analysis will provide more information on PMA-induced megakaryocytic differentiation of K562 cells.

Meg01, M07e, CHRF, and K562 are cell lines derived from the blood and bone marrow of patients with megakaryocytic leukemia [33]. Among these, K562 is a human leukemic cell line that has been widely used as a model of hematopoietic differentiation [12,13]. These cells can be differentiated into hematopoietic lineage cells by differentiation-inducing agents, such as sodium butyrate, hemin, retinoic acid, DMSO, and PMA [14]. Among these agents, PMA is the most potent agent that has been used to differentiate K562 cells into a megakaryocytic cell lineage, demonstrating that the K562 cell line is a common progenitor model for megakaryocytes [15,16]. The concentration of PMA for K562 differentiation into megakaryocytes has varied up to 10 nM, but high concentrations can induce cell apoptosis [12,15,16]. In this study, the differentiation of K562 cells into megakaryocytes following 1 nM PMA treatment was confirmed by cytological examination based on cell morphology and qPCR based on marker gene expression. PMA-induced megakaryocytes exhibited increased size and multilobed nuclei with polyploidization, increased GATA1 and GATA2 expression, and decreased CD34 and *c-KIT* expression. Although CD41 and *GP1B* are important markers for the late-stage differentiation of megakaryocytes [23], their expressions were not upregulated, as analyzed by microarray and qPCR. CD41 and *GP1B* expressions were promoted by the transcription factor GATA [34,35]. Although CD41 and *GP1B* mRNA expressions were not upregulated at the time of observation, GATA1 and GATA2 expressions were significantly increased in megakaryocytes, accompanied by morphological changes in megakaryocytic features. A previous study investigating genetic changes during primary mouse megakaryocyte differentiation showed that the expressions of GATA1 and GATA2, which were high in the early stage of maturation, decreased toward the later stage, whereas the opposite was the case for CD41 and CD61 [36]. Interestingly, although DMSO was used as a control agent for differentiation, DMSO-treated cells did not exhibit adhesive properties, a primary characteristic of megakaryocytes. Moreover, the expressions of a key megakaryocyte marker gene GATA2 and a lymphoid-primed progenitor marker CD34 were not altered in DMSO-treated cells compared with PMA-induced megakaryocytes. These results indicated that PMA is a more potent inducer than DMSO of K562 cell differentiation into megakaryocytes. Mature megakaryocytes express genes, such as *ITGAM*, that can be used to predict the function of the activators of the inflammatory response, and most completely matured cells express genes such as *ITGB3*, also known as CD61, which is required for platelet production [37]. Consistent with this report, microarray analysis showed that *ITGAM* was upregulated and *ITGB3* was downregulated during K562 cell differentiation into megakaryocytes. Furthermore, it has been reported that CD41, *GP1B*, and *ITGB3* are required for glycoprotein signaling for platelet production rather than megakaryocyte maturation [37,38], and CD41 and *GP1B* expressions are somewhat controversial in PMA-differentiated K562 cells [39,40]. According to the microarray analysis, the expressions of *PF4* and *PPBP*, which are mainly released in large amounts from platelets [41], remained unchanged and reduced in PMA-induced megakaryocytes, respectively. *UNC5B* is known to inhibit cell proliferation by inducing cell cycle arrest [42], and *ANXA6* has been reported to induce cell cycle arrest in gastric cancer [43]. The expressions of *UNC5B* and *ANXA6* were increased in PMA-induced megakaryocytes compared to K562 cells based on microarray data, suggesting that cell proliferation was reduced in megakaryocytic differentiation of K562s. Finally, these results indicate that the PMA-induced megakaryocytes used in this study were well differentiated from K562 cells into late-stage megakaryocytes with granule formation but not into the most terminally mature cells for platelet production. Moreover, in the inflammatory response-related genes, anti-inflammatory factors, such as *ITGAM*, *LYZ*, and *P2RX7*, were upregulated, whereas proinflammatory factors, such as *IL1A*, *IL1B*, *TLR5*, *TLR6*, *CCL2*, CD36, CD44, and *CXCL8*, were downregulated. These results indicate that mature megakaryocytes positively prevent inflammation before final maturation for platelet production.

*FN1* was a major gene highly expressed by K562 cell differentiation into megakaryocytes and was included in all five selected functional categories. The expression of *FN1* showed an increased pattern in PMA-induced megakaryocytes at day 4 and significantly increased at day 7 compared to K562 cells (Appendix A). *FN1* is a subunit of fibronectin, a glycoprotein constituting the extracellular matrix [44]. *FN1* promotes cancer cell proliferation, survival, migration, and invasion through focal adhesion kinase activation [45] and stimulates angiogenesis by exhibiting proangiogenic effects associated with AKT signaling [46,47]. *VEGFA* was also a key gene increased in megakaryocytes, included in most selected functional categories. *VEGFA* plays a role in angiogenesis and vasculogenesis and promotes cell migration. It is expressed and secreted by various blood cells, including megakaryocytes, and is required for megakaryocyte maturation through either an autocrine or paracrine mechanism [48]. In addition, *VEGFA* promotes the migration of monocytes and endothelial cells through interaction with various extracellular ligands and induces the migration and invasion of cancer cells [49]. As megakaryocytes are an important reservoir of angiogenic factors [25,32], the increased expressions of *FN1* and *VEGFA* during K562 cell differentiation into megakaryocytes indicate that megakaryocytes promote their expression to strengthen the extracellular matrix and induce angiogenesis.

Megakaryocytes regulate bone metabolism and bone tissue healing by directly interacting with bone cells or indirectly by releasing megakaryocyte-secreted factors [31,50]. These processes not only occur in the extracellular matrix of skeletal tissues but are also correlated with alterations in the extracellular matrix components of connective tissues. Moreover, the local change of extracellular matrix components contributes to the regulation of angiogenesis [51] and plays a role in directing bone remodeling [52]. Interestingly, the expression of *SERPINF1*, included in the extracellular matrix category, was highly increased in megakaryocytes. *SERPINF1*, a multifunctional secreted protein, has anti-angiogenic functions via regulation of VEGF signaling [53]. However, several studies have shown that *SERPINF1* promotes mesenchymal stem cell differentiation and increases bone matrix mineralization by osteoblasts [54,55]. These reports suggest that mature megakaryocytes indirectly regulate bone metabolism. We recently identified megakaryocyte-secreted factors in PMA-induced megakaryocytes and reported that the megakaryocyte-derived factor *S100P* promotes osteoclast differentiation and bone resorption, evidencing the indirect role of megakaryocytes in bone remodeling [31]. In addition to *S100P*, *IDH1*, *P2RX7*, and *BST2*, involved in secretion, play a role in skeletal tissue. *IDH1* mutation inhibits osteogenic differentiation [56], and *P2RX7* mutation causes osteopenia exhibiting deficient periosteal bone formation [57,58]. *BST2* increases osteogenic differentiation by regulating the BMP2 signaling pathway [59]. These factors were significantly increased in mature megakaryocytes, suggesting that megakaryocyte-derived factors regulate bone remodeling and metabolism. Moreover, megakaryocytes express fibronectin and collagen and contribute to bone marrow extracellular matrix homeostasis [30]. Consistent with this report, *FN1* was highly expressed in the five selected functional categories classified by K562 cell differentiation into megakaryocytes. On the other hand, MMP1 and TIMP1 play a role in regulating extracellular matrix composition and wound healing [60]. CILP is expressed in articular cartilage, and its expression is increased in early osteoarthritic cartilage or during cartilage degeneration [61,62]. TFPI2 and A2M inhibit proteases, thereby regulating matrix degradation [63,64,65]. These genes involved in the extracellular matrix category were downregulated in mature megakaryocytes, indicating that megakaryocytes play a role in modulating the extracellular matrix environment.

In conclusion, this is the first study to investigate DEGs during K562 leukemia cell differentiation into megakaryocytes using an Affymetrix GeneChip microarray and to analyze the protein–protein interactions translated from DEGs based on the STRING database. Megakaryocytes have roles in physiological and pathological conditions, including prevention of inflammation before final maturation for platelet production, promotion of angiogenesis, and regulation of the extracellular matrix environment in the body. Further studies using knockdown experiments will be required to confirm the roles of the important genes identified here through the microarray analysis. Additional studies will also be needed to compare the gene expression profiles in normal megakaryocytes with the current results and to validate the expressions of genes in normal megakaryocytes or other cell lines, such as Meg-01, M07e, or CHRF. Although this work lacks an assessment of translational relevance, further studies with more robust analyses or comparisons with other datasets may support the findings of this study. Finally, our results provide bioinformatic-based information for further studies to elucidate the multifunctionality of megakaryocytes.

## 4. Materials and Methods

### 4.1. Cell Culture

The human leukemia cell line K562 (ATCC, Manassas, VA, USA) was used to generate megakaryocytes [12]. K562 cells were cultured in RPMI 1640 medium (HyClone, Logan, UT, USA) containing 10% fetal bovine serum (HyClone), 10 U/mL penicillin, and 10 μg/mL streptomycin (Gibco, Grand Island, NY, USA) and maintained in culture for up to passage number 8. The K562 cell suspension has been recommended to split for a subculture or use for megakaryocyte differentiation at the time point when cells form aggregates. K562 cells were seeded at 3 × 10^5^ cells/mL, and cell cultures were started in differentiation medium when they reached a density of 1.5 × 10^6^ cells/mL. The megakaryocytic differentiation of K562 cells was induced by treating them with 1 nM PMA (Sigma-Aldrich, St. Louis, MO, USA) or 0.1% DMSO for 5 and 7 days. The differentiation medium supplemented with inducing agents was replaced every 2–3 days, and each time the cells were collected by centrifugation and transferred into a new culture dish.

### 4.2. Cell Morphology and Polyploidy Analysis

Differentiated cell morphology was determined by cytological staining using Hemacolor Rapid staining of blood smear kit (Merck, Darmstadt, Germany) following the manufacturer’s instructions. Briefly, after culturing K562 cells with PMA for 7 days, the cells were centrifuged and collected. Next, the fixing solution was added to the precipitated cell pellet for 30 s. The fixed cells were then smeared onto a slide glass and air-dried at room temperature. After confirming that the slide glass was completely dried, the cells were stained with color reagent red for 15 s and washed with tap water. Subsequently, they were stained with color reagent blue for 5 s and washed with running tap water. After that, the cell morphology was analyzed at 40x magnification under a DM1000 LED light microscope (Leica Microsystems, Wetzlar, Germany), and cell diameters were measured using i-Solution image analysis software (IMT i-Solution, Daejeon, Korea). Cell adhesion was observed in the attached cells by Hemacolor Rapid staining after culturing K562 cells with 1 nM PMA or 0.1% DMSO for 7 days. Cell adhesion before and after staining was visualized under a CKX41 inverted microscope (Olympus Co., Tokyo, Japan) and S8 APO (Leica Microsystems), respectively. FACS analysis was performed for megakaryocyte ploidy analysis. Cells cultured on 100 mm culture plates were harvested and fixed with ice-cold ethanol. After removing the supernatant by centrifugation, the cells were washed with phosphate-buffered saline at 4 °C. Each 10^6^ cells were then stained with 500 μL of propidium iodide staining solution (BD Biosciences, Franklin Lakes, NJ, USA) for 30 min in the dark and analyzed using FACS Calibur (BD Biosciences).

### 4.3. RNA Extraction and qPCR

Total RNA was extracted from cultured cells using TRIzol reagent (Sigma-Aldrich, St. Louis, MO, USA). Reverse transcription was performed using reverse transcriptase premix (ELPIS-Biotech, Daejeon, Korea) with 1 µg of total RNA to synthesize cDNA. qPCR was performed using Power SYBR Green PCR master mix (Applied Biosystems, Waltham, MA, USA) on a StepOnePlus real-time PCR system (Applied Biosystems) as follows: initial denaturation at 95 °C for 5 min; 45 cycles of amplification with denaturation at 95 °C for 30 s, annealing at 64 °C for 30 s, and extension at 72 °C for 1 min; 1 cycle for melting curve analysis at 95 °C for 5 s, 65 °C for 1 min, and 97 °C continuous; and a final cooling step at 40 °C for 30 s. The results were analyzed using the comparative cycle threshold (*C_T_*) method. The primer sets used for qPCR are listed in Appendix A.

### 4.4. Microarray

Gene expression was analyzed using the GeneChip^®^ Human Gene 2.0 ST array (Affymetrix, Santa Clara, CA, USA) containing > 418,000 exon-level probe sets and >48,000 gene-level probe sets. It was examined by 40,716 well-established and annotated RefSeq transcripts using over 1,350,000 different probes. Array configuration was based on February 2012 RefSeq (NCBI build 51), Ensembl (Release 65), and lncRNAdb (http://lncrnadb.com, assessed on 29 April 2019).

Affymetrix GeneChip processes were performed following the manufacturer’s instructions. Briefly, cDNA was synthesized from total RNA with a high RNA integrity number (RIN) > 9.0 using a GeneChip^®^ WT PLUS Reagent Kit including SuperScript II reverse transcriptase, T7-(N)6 primer, and Random primer. Subsequently, single-stranded cDNA was isolated using a WT Amplification Kit Module 2 (Affymetrix) and fragmented by enzymatic treatment at 37 °C for 1 h with a GeneChip^®^ WT Terminal Labeling Kit (Affymetrix). The digested cDNA was labeled with biotin and then injected into GeneChip at 45 °C for 17 h, followed by hybridization. After hybridization, the GeneChips were stained and washed in Fluidics Station 450 (Affymetrix) using a GeneChip^®^ Hybridization, Wash, and Stain Kit. Images were scanned from each array using a GeneChip^®^ Scanner 3000 7G (Affymetrix), and the signal intensity was calculated for determining gene expression levels using Expression Console™ software, Version 1.4.1 (Affymetrix). The heat map was presented using ExDEGA v1.6.5 (Ebiogen, Seoul, South Korea) and created through fold change based on the z-score.
$$Z-\mathrm{score}=\frac{\left\{\mathrm{Normalized}\;\mathrm{data}\;\left(\log10\right)-\mathrm{Average}\;\mathrm{of}\;\mathrm{Normalized}\;\mathrm{data}\;\left(\log10\right)\right\}}{\mathrm{Standard}\;\mathrm{deviation}\;\mathrm{of}\;\mathrm{Normalized}\;\mathrm{data}\;\left(\log10\right)}$$

### 4.5. STRING Database Analysis

The STRING database (http://string-db.org, version 11.5 assessed on 1 November 2021) was used to analyze the interactions between proteins constituting each selected category. The STRING database can be used to assess the type and strength of the interactions between proteins. The full STRING network was used to analyze the direct (physical) and indirect (functional) interactions of proteins. The thickness of the connecting line between proteins depended on the number of evidence for interactions. The sources of evidence for protein interactions included text mining, experiments, databases, co-expression, neighborhood, gene fusion, and co-occurrence.

### 4.6. Statistical Analysis

Cell culture experiments were performed independently at least three times. RNA samples were collected from each cultured group, and gene expression analysis was performed three times independently. All data are presented as the mean ± standard deviation. The Student’s *t*-test was used to analyze differences between values and a *p*-value < 0.05 was considered statistically significant.

## References

## Figures and Tables

**Figure 1 ijms-23-04221-f001:**
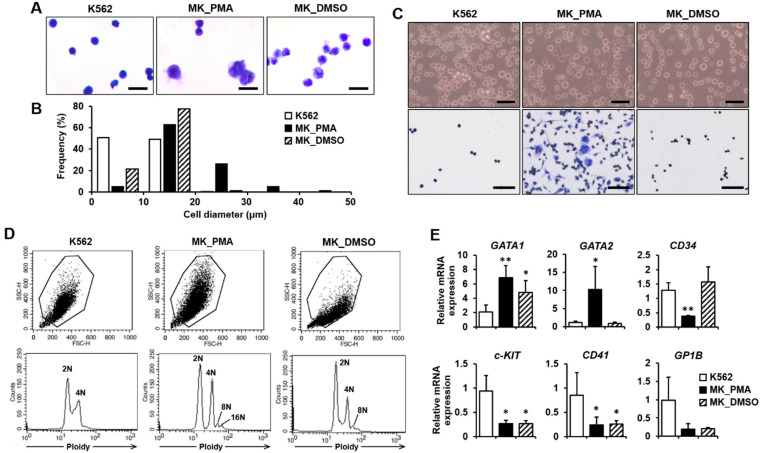
Verification of K562 cell differentiation into megakaryocytes. (**A**,**B**) The morphologies and diameters of K562 cells (K562s), PMA-induced megakaryocytes (MK_PMAs), and DMSO-treated megakaryocytes (MK_DMSOs) were comparatively analyzed through cytological staining (**A**) and size quantification (**B**). K562 cells were treated with PMA or DMSO for 7 days. Cell size was classified into 10 μm increments ranging from 0 to 50 μm in diameter. A total of 626 K562s, 401 MK_PMAs, and 464 MK_DMSOs were quantified by the percentage of frequency using the i-Solution image analysis program. Scale bar = 25 μm. (**C**) Verification of K562 cell differentiation into megakaryocytes by cell adherence. Adhesive properties to the culture plate were verified before and after Hemacolor Rapid staining, as shown in the upper and lower panels, respectively. Scale bar = 100 μm. (**D**) Representative data showing ploidy of megakaryocytes after 7 days of differentiation with PMA. MK_PMAs were compared with K562s or MK_DMSOs. (**E**) mRNA levels of GATA1, GATA2, CD34, *c-KIT*, CD41, and *GP1B* were evaluated using qPCR in MK_PMAs or MK_DMSOs and compared with those in K562 cells. The relative mRNA expression level was plotted against gene expression level for K562 cells. The results are presented as the mean ± standard deviation of three independent experiments. *, *p* < 0.05; **, *p* < 0.01 versus K562 cells.

**Figure 2 ijms-23-04221-f002:**
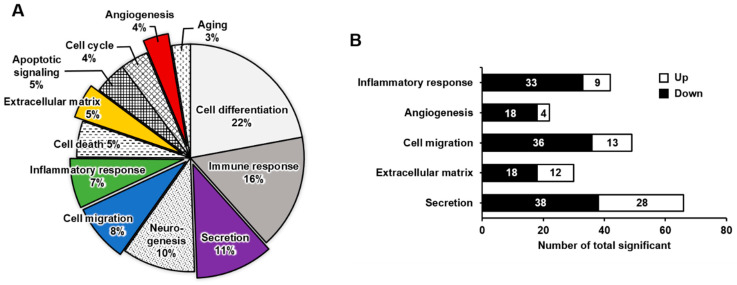
Upregulated or downregulated gene expression patterns associated with K562 cell differentiation into megakaryocytes, presented by gene category. (**A**) Pie chart illustrating the functional classification of differentially expressed genes between K562 cells and megakaryocytes. The ratio of genes with >2-fold change and a *p*-value < 0.05 is indicated. (**B**) The number of upregulated or downregulated genes in five selected clusters: inflammatory response, angiogenesis, cell migration, extracellular matrix, and secretion.

**Figure 3 ijms-23-04221-f003:**
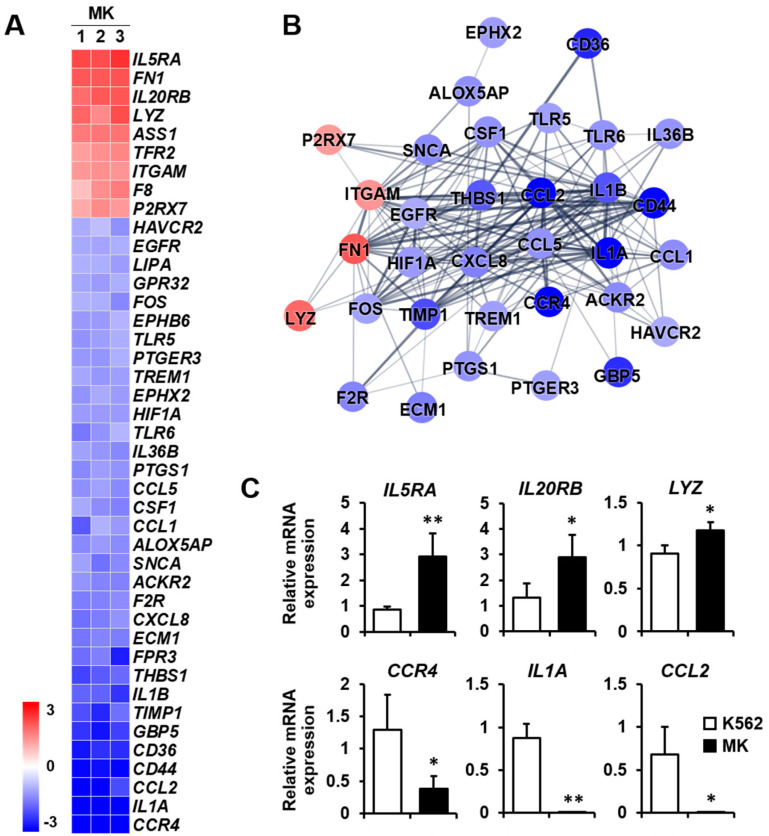
Expression levels by normalized z-scores of genes included in the inflammatory response cluster and the relationship between constituent genes. (**A**) Heat map of genes differentially expressed in megakaryocytes (MK) compared with K562 cells based on the microarray analysis. The red color indicates upregulation and the blue color indicates downregulation. The degrees of colors represent the expression levels of the genes. (**B**) Protein–protein interaction analysis using the STRING database. The red and blue colors denote upregulation and downregulation, respectively. The higher the confidence in the interaction, the greater the line thickness between the proteins. Members without interaction with other genes were excluded from the STRING database analysis. (**C**) Validation of the mRNA expression of selected genes (*IL5RA*, *IL20RB*, *LYZ*, *CCR4*, *IL1A*, and *CCL2*) differentially expressed in MK by qPCR. Relative expression levels were plotted against gene expression levels in K562 cells. Data are presented as the mean ± standard deviation of three independent experiments. Please see abbreviations for the full names of genes. *, *p* < 0.05; **, *p* < 0.01 versus K562 cells.

**Figure 4 ijms-23-04221-f004:**
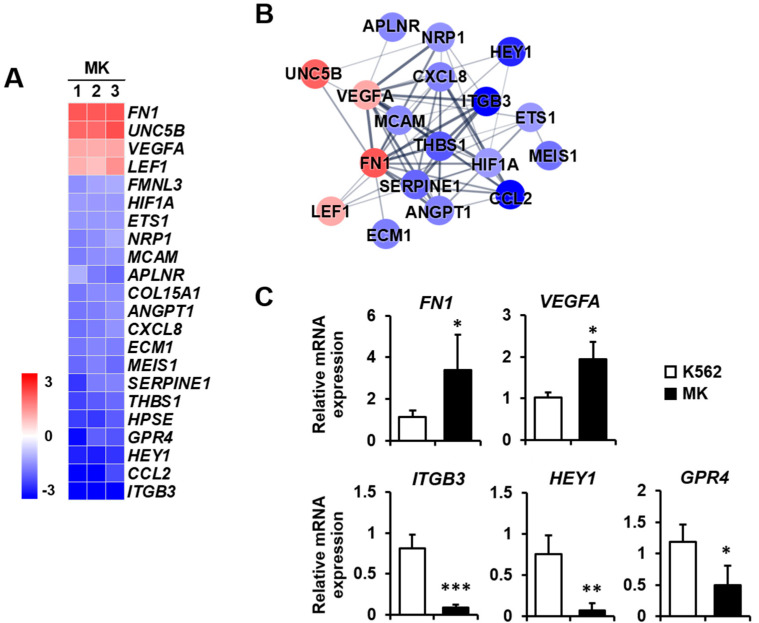
Expression levels by z-scores of genes included in the angiogenesis cluster and the relationship between constituent genes. (**A**) Heat map of genes differentially expressed in MK compared with K562 cells based on the microarray analysis. The red color indicates upregulation and the blue color indicates downregulation. The degrees of colors represent the expression levels of the genes. (**B**) Protein–protein interaction analysis using the STRING database. The red and blue colors denote upregulation and downregulation, respectively. The higher the confidence in the interaction, the greater the line thickness between the proteins. Members without interaction with other genes were excluded from the STRING database analysis. (**C**) Validation of the mRNA expression of selected genes (*FN1*, *VEGFA*, *ITGB3*, *HEY1*, and *GPR4*) differentially expressed in MK by qPCR. Relative expression levels were plotted against gene expression levels in K562 cells. Data are presented as the mean ± standard deviation of three independent experiments. Please see abbreviations for the full names of genes. *, *p* < 0.05; **, *p* < 0.01; ***, *p* < 0.001 versus K562 cells.

**Figure 5 ijms-23-04221-f005:**
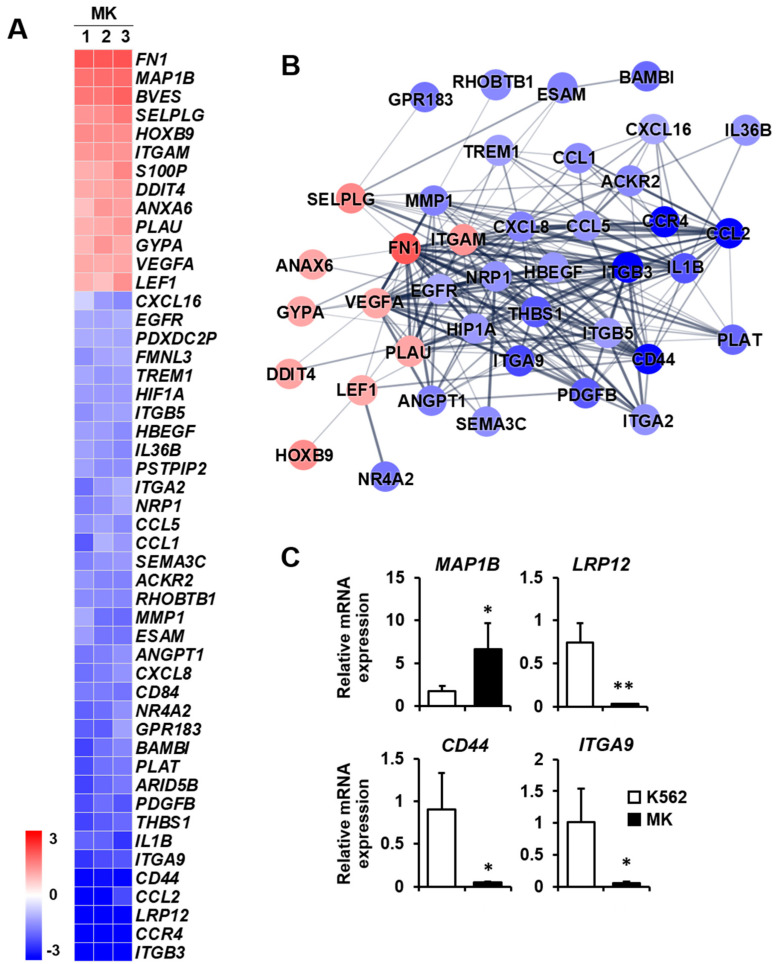
Expression levels by z-scores of genes included in the cell migration cluster and the relationship between constituent genes. (**A**) Heat map of genes differentially expressed in megakaryocytes (MKs) compared with K562 cells based on the microarray analysis. The red color indicates upregulation and the blue color indicates downregulation. The degrees of colors represent the expression levels of the genes. (**B**) Protein–protein interaction analysis using the STRING database. The red and blue colors denote upregulation and downregulation, respectively. The higher the confidence in the interaction, the greater the line thickness between the proteins. Members without interaction with other genes were excluded from the STRING database analysis. (**C**) Validation of the mRNA expression of selected genes (*MAP1B*, *LRP12*, CD44, and *ITGA9*) differentially expressed in MKs by qPCR. Relative expression levels were plotted against gene expression levels in K562 cells. Data are presented as the mean ± standard deviation of three independent experiments. Please see abbreviations for the full names of genes. *, *p* < 0.05; **, *p* < 0.01 versus K562 cells.

**Figure 6 ijms-23-04221-f006:**
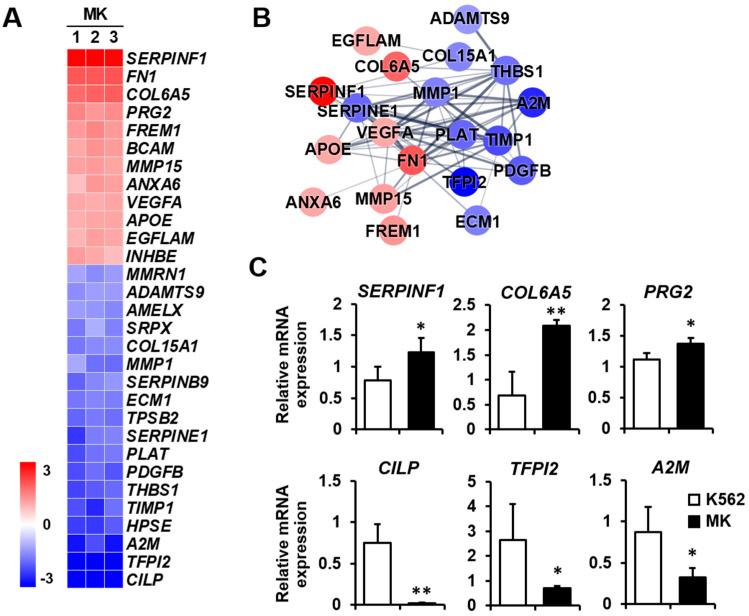
Expression levels by z-scores of genes included in the extracellular matrix cluster and the relationship between constituent genes. (**A**) Heat map of genes differentially expressed in megakaryocytes (MKs) compared with K562 cells based on the microarray analysis. The red color indicates upregulation and the blue color indicates downregulation. The degrees of colors represent the expression levels of the genes. (**B**) Protein–protein interaction analysis using the STRING database. The red and blue colors denote upregulation and downregulation, respectively. The higher the confidence in the interaction, the greater the line thickness between the proteins. Members without interaction with other genes were excluded from the STRING database analysis. (**C**) Validation of the mRNA expression of selected genes (*SERPINF1*, *COL6A5*, *PRG2*, *CILP*, *TFPI2*, and *A2M*) differentially expressed in MKs by qPCR. Relative expression levels were plotted against gene expression levels in K562 cells. Data are presented as the mean ± standard deviation of three independent experiments. Please see abbreviations for the full names of genes. *, *p* < 0.05; **, *p* < 0.01 versus K562 cells.

**Figure 7 ijms-23-04221-f007:**
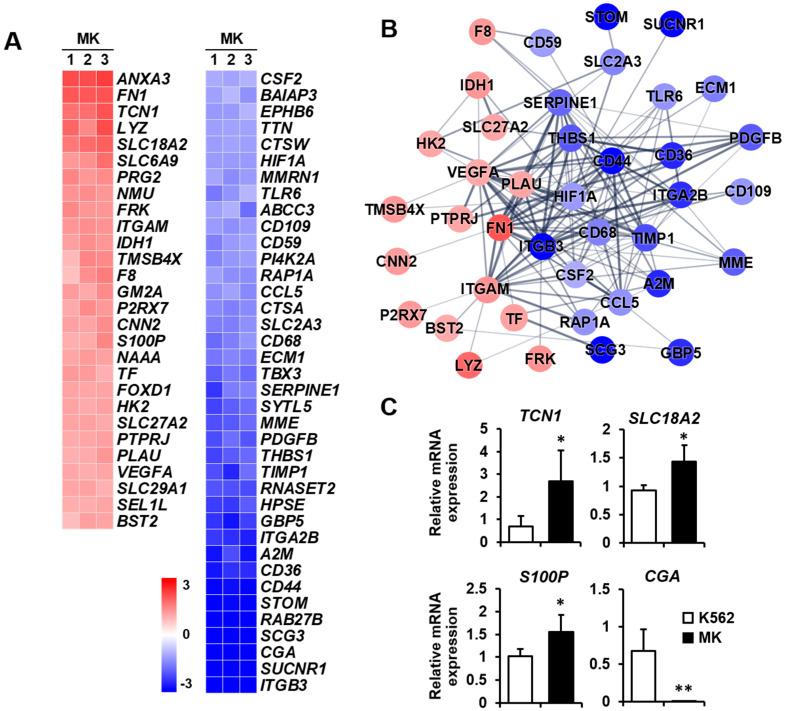
Expression levels by z-scores of genes included in the secretion cluster and the relationship between constituent genes. (**A**) Heat map of genes differentially expressed in megakaryocytes (MKs) compared with K562 cells based on the microarray analysis. The red color indicates upregulation and the blue color indicates downregulation. The degrees of colors represent the expression levels of the genes. (**B**) Protein–protein interaction analysis using the STRING database. The red and blue colors denote upregulation and downregulation, respectively. The higher the confidence in the interaction, the greater the line thickness between the proteins. Members without interaction with other genes were excluded from the STRING database analysis. (**C**) Validation of the mRNA expression of selected genes (*TCN1*, *SLC18A2*, *S100P*, and *CGA*) differentially expressed in MKs by qPCR. Relative expression levels were plotted against gene expression levels in K562 cells. Data are presented as the mean ± standard deviation of three independent experiments. Please see abbreviations for the full names of genes. *, *p* < 0.05; **, *p* < 0.01 versus K562 cells.

## Data Availability

Data can be obtained from the corresponding author on reasonable request.

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
