# Peer review of "Bioinformatics of Differentially Expressed Genes in Phorbol 12-Myristate 13-Acetate-Induced Megakaryocytic Differentiation of K562 Cells by Microarray Analysis"

_ijms, 2022, doi:10.3390/ijms23084221_

Round 1
Reviewer 1 Report
The manuscript is ready for publication
Author Response
Reviewer #1
The manuscript is ready for publication.
>> Thank you for reviewing our manuscript and providing helpful comments on it.
Reviewer 2 Report
In the first review, all reviewers point out the single use of K562 leukemia cells and whether K562 cells with PMA treatment can recapitulate human megakaryocyte differentiation is problematic. Although the authors have addressed some concerns with new data and edited the manuscripts as suggested by reviewers, authors did not clearly address this main issue and there is no normal megakaryocyte analysis. Failure to show CD41 and GP1b expression in this cell line also implicates their incomplete differentiation. These discrepancies significantly hamper the author conclusion.
Author Response
Reviewer 2
In the first review, all reviewers point out the single use of K562 leukemia cells and whether K562 cells with PMA treatment can recapitulate human megakaryocyte differentiation is problematic. Although the authors have addressed some concerns with new data and edited the manuscripts as suggested by reviewers, authors did not clearly address this main issue and there is no normal megakaryocyte analysis. Failure to show CD41 and GP1b expression in this cell line also implicates their incomplete differentiation. These discrepancies significantly hamper the author conclusion.
>> We appreciate these valuable comments and concerns. Based on previous reports, it is known that K562 is a human leukemic cell line that is widely used as a model of hematopoietic differentiation including megakaryocytes, and PMA has been used to differentiate K562 cells into a megakaryocytic cell lineage, showing that K562 cell line is a common progenitor model for megakaryocytes. Although CD41 and GP1B are important markers for the late-stage differentiation of megakaryocytes, their expression was not upregulated as analyzed through microarray and qPCR. Although CD41 and GP1B mRNA expressions were not upregulated at the time of observation, the expressions of GATA1 and GATA2, which promote CD41 and GP1B expressions, were significantly increased in megakaryocytes, accompanied by morphological changes in megakaryocytic features. In addition, during K562 cell differentiation into megakaryocytes, ITGAM, which predicts the function of the activators of the inflammatory response, was upregulated, and ITGB3, which is required for platelet production, was downregulated. Finally, these results indicate that the PMA-induced megakaryocytes used in this study were well-differentiated from K562 cells into late-stage megakaryocytes with granule formation but not into the most finally mature cells for platelet production. As the reviewer’s comment, additional studies will be needed to compare the gene expression profiles in normal megakaryocytes with the current results and to validate gene expressions in other megakaryocytic cell lines. We presented the results and described these points in the Discussion section with references.
Reviewer 3 Report
In this manuscript, the authors analyzed the differentiation of K562 cells into megakaryocytic induced by PMA, at the transcriptomic level. The work presents a useful resource for identifying megakaryocyte-specific genes during maturation, using high-throughput analysis of genetic information. Overall, the manuscript is interesting but lacks quality because of many issues that I identified here below.
In vitro cultures
- What was the passage, and the exact number of cells used in the different assays? How were the cells maintained? Was it a single PMS/DMSO treatment for 7 days?
Results/Discussion
- Figure 1- Megakaryocytic differentiation process is characterized by changes in cell morphology, adhesive properties; these changes should have been accessed across the 7 days maturation period.
Pictures provided have very low resolution: for instance, it is impossible to visualize the “multilobed nuclei” or discriminate cells with different diameters.
Figure 1A is composed of 2 plots and we should categorize as “A” and “A’” (at least).
Figure 1C: FACS analysis must be detailed in material and methods! As well as the microscopic analysis…
Overall, why did the authors only provide results for the MK_DMSO group in Figure 1D (qPCR); such control should be included in all the analyses.
Could the authors comment on what could explain the lack of CD41 expression? What about CD61? What do the bars of Figure 1D mean, +/-SEM? What is the N for each assay?
Microarray analysis
- Overall, it is difficult to follow the analysis that the authors presented here.
I would like to suggest that after describing the total number of expressed genes, authors should first discriminate the number of DEGs; and amongst them the number of up- and down-regulated DEGs (highlighting some examples, such as the top5-10 up- or down-regulated DEGs, etc.). In addition, a correlation between qPCR data (Figure 1) should be highlighted.
Afterward, through a PCA (for instance), authors should clearly discriminate the cluster represented in Figure 2 and characterize those clusters: number of DEGs; up- and down-regulated DEGs. It would be helpful a heatmap showing the pattern of expression for each cluster and a plot showing the pattern of expression for specific genes characterizing each cluster.
Why and how did the authors choose to characterize only these specific 5 clusters (Figures 3-7)?
- What is the genetic profile (signature) of the MK_PMA cells, at day 7 of maturation? Are cells completely maturated into megakaryocytes? Or still under differentiation? Authors should clearly provide plots (heatmap/PCA/tSNE) showing this signature/profile. How different is this signature from K562 cells? And from MK_DMSO (why didn’t the authors perform this group?)? These differences should be shown and are crucial for the relevance of this work. How novel is this information?
- Why did the authors choose to analyze only one time-point? And why day 7?
- Figure 2B- it would be helpful if authors would add the number of DEGs to the bars.
- Figures 3-7: authors always associate a heatmap characterizing a specific cluster with a protein interaction chart based on the hypothetical functional enrichment analysis. The validation of some genes characterizing each cluster is crucial, otherwise what is the purpose of such an analysis? What is the relevance of the genes depicted in the heatmap/interaction map? Authors should try to complement with a more robust informatics analysis, by qPCR, FACS, or IF.
The megakaryocytic differentiation process is characterized by changes in cell morphology, adhesive properties, and expression of markers associated with megakaryocytes as well as cell growth arrest. Are the authors able to correlate events such as mitogen-activated protein kinase activation, cell cycle arrest, or expression of some transcription factors with the genetic signature characterizing the different clusters described here?
- Of note, the selectivity of megakaryocyte genes might be debatable compared to (single-cell) RNA sequencing experiments. Because there will be considerable overlap with gene expression of other cell types. Could the authors comment?
- What is the relevance of FN1 for megakaryocytes? Is its pattern of expression different during maturation? How different is its level of expression across the different clusters analyzed?
- The work lacks translational relevance assessment, which should be complemented with more robust analysis and comparison with other datasets.
- Methods are not well described and lack detailed information, including methods presented in the Results section and statistical analysis of the microarray data (i.e., z-scores, estimated PFP, etc.). How the authors did build the heatmaps (R packages?).
- Figure legends must be edited, they lack detailed information on the assays represented by the plots.
- Parts of the Introduction section should be rather placed in the Discussion section. Both sections need more details: megakaryocytic differentiation of K562 cells induced by PMA needs a proper and detailed introduction; results need a comprehensive discussion.
- Authors should proofread the manuscript (including the title), minimize typos, grammatical errors, and provide a list of abbreviations.
Round 2
Reviewer 2 Report
Authors addressed the most concerns with new data and properly discussed about the strength and weakness of current research in revised manuscript.
This manuscript is a resubmission of an earlier submission. The following is a list of the peer review reports and author responses from that submission.
Round 1
Reviewer 1 Report
In the current study the authors present data about gene expression changes during megakaryocyte maturation. Authors traced the gene expression changes during K562 cell differentiation into human megakaryocytes using Affymetrix microarray and screened genes showing distinct changes in the MK differentiation. Authors found gene categories and the multifunctionality of human megakaryocytes. Overall, the topic and approach are interesting, however there are several concerns about the MK markers and validation of key gene expression in normal human MK cells.
In Figure 1, authors tested GATA1 and 2 expression for human megakaryocyte differentiation. Authors should test mature MK markers including platelet factor 4 (PF4) and pro-platelet basic protein (PPBP).
The major limitation of this study is to use K562 leukemia cells for human megakaryocyte differentiation. It has been reported that TPO alone can promote megakaryocyte differentiation of BM (or CB)-derived CD34+ cells. It will be much more informative if authors use normal megakaryocytes or in vitro differentiated MKs from normal CD34+ cells and to compare the gene expression profiles with current results.
Authors showed the list of genes upregulated in K562 MK differentiation. Authors need to validate whether normal MK cells indeed express those top listed genes.
Author Response
Reviewer #1
In the current study the authors present data about gene expression changes during megakaryocyte maturation. Authors traced the gene expression changes during K562 cell differentiation into human megakaryocytes using Affymetrix microarray and screened genes showing distinct changes in the MK differentiation. Authors found gene categories and the multifunctionality of human megakaryocytes. Overall, the topic and approach are interesting, however there are several concerns about the MK markers and validation of key gene expression in normal human MK cells.
In Figure 1, authors tested GATA1 and 2 expression for human megakaryocyte differentiation. Authors should test mature MK markers including platelet factor 4 (PF4) and pro-platelet basic protein (PPBP).
>> We appreciate this valuable comment. To verify true differentiation into megakaryocytes, we conducted qPCR to observe the expression of megakaryocyte markers, such as GATA1, GATA2, CD34, and c-KIT, and we analyzed megakaryocyte polyploidization, cell size, and cell adherence, which are major features of megakaryocytes. These results were presented in Figure 1 and Supplementary Figure S1. However, the expression of CD41 and GP1B, another markers for late-stage differentiation of megakaryocytes, was not upregulated. CD41 and GP1B expression was promoted by a transcription factor GATA [Ludlow et al., 1996; Jackers et al., 2004]. Although CD41 and GP1B mRNA expression was not upregulated at the time point which we observed, GATA1 and GATA2 expression was significantly increased in megakaryocytes, accompanied by enlarged cell size and multilobed nucleus. Particularly, it has been reported that CD41, GP1B, and ITGB3 are needed for glycoprotein signaling for platelet production rather than megakaryocyte maturation [Sun et al., 2019; Lentaigne et al., 2016]. Additionally, there are some reports that CD41 and GP1B expression was detected in K562 cells that have not differentiated into megakaryocytes or somewhat controversial in PMA-differentiated K562 cells [Rouillard et al., 2016; Conde et al., 2016]. Although we could not test PF4 and PPBP expression due to time limitation for revision, we could find that PF4 expression remained unchanged and PPBP expression was reduced in PMA-induced megakaryocytes from the microarray datasets of Supplementary Figure S2. PF4 and PPBP are known to be mainly released in large amounts from platelets [Supernat et al., 2021]. Finally, all the results we observed indicated that PMA-induced megakaryocytes used for this study were well differentiated from K562 cells into mature megakaryocytes as late-stage megakaryocytes with granule formation, not the most finally mature cells for platelet production. We have described this in the Discussion section (Lines 273-290) with References below.
- Ludlow, L.B.; Schick, B.P.; Budarf, M.L.; Driscoll, D.A.; Zackai, E.H.; Cohen, A.; Konkle, B.A. Identification of a mutation in a GATA bindingsite of the platelet glycoprotein Ibbeta promoter resulting in the Bernard-Soulier syndrome. Biol. Chem. 1996, 271, 22076-22080.
- Jackers, P.; Szalai, G.; Moussa, O.; Watson, D.K. Ets-dependent regulation of target gene expression during megakaryopoiesis. Biol. Chem. 2004, 279, 52183-52190.
- Sun, S.; Jin, C.; Li, Y.; Si, J.; Cui, Y.; Rondina, M.T.; Tang, F.; Wang, Q.F. Transcriptional and spatial heterogeneity of mouse megakaryocytes at single-cell resolution. Blood2019, 134, 275.
- Lentaigne, C.;Freson, K.; Laffan, M.A.; Turro, E.; Ouwehand, W.H.; BRIDGE-BPD Consortium and the ThromboGenomics Consortium. Inherited platelet disorders: toward DNA-based diagnosis. Blood 2016, 127, 2814-2823.
- Rouillard, A.D.; Gundersen, G.W.; Fernandez, N.F.; Wang, Z.; Monteiro, C.D.; McDermott, M.G.; Ma'ayan, A. The harmonizome: a collection of processed datasets gathered to serve and mine knowledge about genes and proteins. Database (Oxford). 2016, 2016, baw100.
- Conde, I.; Pabón, D.; Jayo, A.; Lastres, P.; González-Manchón, C. Involvement of ERK1/2, p38 and PI3K in megakaryocytic differentiation of K562 cells. J. Haematol. 2010, 84, 430-440.
- Supernat, A.; PopÄ™da, M.; Pastuszak, K.; Best, M.G.; Grešner, P.; Veld, S.I'.; Siek, B.; Bednarz-Knoll, N.; Rondina, M.T.; Stokowy, T.; Wurdinger, T.; Jassem, J.; Å»aczek, A.J. Transcriptomic landscape of blood platelets in healthy donors. Rep. 2021, 11, 15679.
The major limitation of this study is to use K562 leukemia cells for human megakaryocyte differentiation. It has been reported that TPO alone can promote megakaryocyte differentiation of BM (or CB)-derived CD34+ cells. It will be much more informative if authors use normal megakaryocytes or in vitro differentiated MKs from normal CD34+ cells and to compare the gene expression profiles with current results.
>> We appreciate this valuable comment. Notably, K562 has been reported to be a human leukemic cell line used as a model of hematopoietic differentiation. It can be differentiated into different hematopoietic lineage cells by various differentiation-inducing agents, such as sodium butyrate, hemin, retinoic acid, DMSO, and PMA [Sutherland et al., 1986]. Among these agents, the potent agent PMA has been used for K562 cells to differentiate into megakaryocytic cell lineage, showing that K562 cell line is a common progenitor model for megakaryocytes [Huo et al., 2006; Kim et al., 2001]. Based on previous reports, therefore, we used K562 cells to differentiate megakaryocytes using PMA. We have described this in the Discussion section (Lines 264-269) with References below. Unfortunately, we have not prepared normal megakaryocytes or in vitro differentiated megakaryocytes from normal CD34+ cells. However, further studies will be helpful to compare the gene expression profiles with current results and we have described the necessity of further study in the Discussion section (Lines 345-346).
- Sutherland, J.A.; Turner, A.R.;Mannoni, P.; McGann, L.E.; Turc, J.M. Differentiation of K562 leukemia cells along erythroid, macrophage, and megakaryocyte lineages. Biol. Response Mod. 1986, 5, 250-262.
- Huo, X.F.; Yu, J.; Peng, H.; Du, Z.W.; Liu, X.L.; Ma, Y.N.; Zhang, X.; Zhang, Y.; Zhao, H.L.; Zhang, J.W. Differential expression changes in K562 cells during the hemin-induced erythroid differentiation and the phorbol myristate acetate (PMA)-induced megakaryocytic differentiation. Cell. Biochem. 2006, 292, 155-167.
- Kim, K.W.; Kim, S.H.; Lee, E.Y.; Kim, N.D.; Kang, H.S.; Kim, H.D.; Chung, B.S.; Kang, C.D. Extracellular signal-regulated kinase/90-KDA ribosomal S6 kinase/nuclear factor-kappa B pathway mediates phorbol 12-myristate 13-acetate-induced megakaryocytic differentiation of K562 cells. Biol. Chem. 2001, 276, 13186-13191.
Authors showed the list of genes upregulated in K562 MK differentiation. Authors need to validate whether normal MK cells indeed express those top listed genes.
>> We appreciate the reviewer’s suggestion. Unfortunately, as mentioned to the upper response, we have not prepared normal megakaryocytes or in vitro differentiated megakaryocytes from normal CD34+ cells. However, further studies will be needed to validate the top listed gene expression in normal megakaryocytes. We have discussed the implications of this further study in the Discussion section (Lines 346-347).

Reviewer 2 Report
In this manuscript Lee et al., has provided gene expression analysis of differentiating K562 cell lines into the megakaryocyte lineage using microarrays. In recent times multiple groups have established protocols for generating in-vitro platelet production by differentiating megakaryocytes. However, the yield of artificially generated platelets have not been sufficient for transfusion in patients. Thus, the data sets provided in this manuscripts may prove important in understanding megakaryocyte production. However, there are multiple flaws that the authors need to address prior to publicaiton.
- Why was microarray used for this study when RNA-seq data is more reliable?
- K562 cells were differentiated with PMA. However, did the cells truly differentiate into megakaryocyte lineage? The authors are requested to provide ploidy analysis and CD41 and GP1b expression levels the differentiated cells used for microarray analysis
- The authors need to confirm roles of important genes which were identified in the microarray analysis in the differentiation process through knock downs
- CD41 and GP1b are very import megakaryocyte markers. It is surprising that these genes are not up-regulated in the datasets. The authors need to justify this phenomenon
Author Response
Reviewer #2
In this manuscript Lee et al., has provided gene expression analysis of differentiating K562 cell lines into the megakaryocyte lineage using microarrays. In recent times multiple groups have established protocols for generating in-vitro platelet production by differentiating megakaryocytes. However, the yield of artificially generated platelets have not been sufficient for transfusion in patients. Thus, the data sets provided in this manuscripts may prove important in understanding megakaryocyte production. However, there are multiple flaws that the authors need to address prior to publication.
- Why was microarray used for this study when RNA-seq data is more reliable?
>> Thank you for this insightful question. Based on the reviewer’s comment, RNA-seq has the advantage of identifying single-nucleotide polymorphisms because it can perform a sequence-based analysis. Additionally, RNA-seq is more sensitive and accurate than microarrays. However, there are many limitations, such as large amounts of data, the complexity of analysis, non-optimized protocol, and almost importantly, high cost. They should be overcome to become the predominantly used method for transcriptome analysis. Alternatively, microarrays are still used in clinical diagnostic testing for diseases because they are a well-established process and cost-effective for large-scale studies. Therefore, we choose microarrays rather than RNA-seq, as a more universal and accessible technology, to demonstrate the general possibility of megakaryocyte multifunctionality in terms of bioinformatics. In the Introduction section (Lines 52-60), we have described this point with References below.
- Rodriguez-Esteban, R.; Jiang, X. Differential gene expression in disease: a comparison between high-throughput studies and the literature. BMC Med. Genomics 2017, 10, 59.
- Zhao, S.; Fung-Leung, W.P.; Bittner, A.; Ngo, K.; Liu, X. Comparison of RNA-Seq and microarray in transcriptome profiling of activated T cells. PLoS ONE 2014, 9, e78644.
- Lemuth, K.; Rupp, S. Microarrays as research tools and diagnostic devices. RNA and DNA Diagnostics, Springer: Switzerland, 2015; 259-280.
- Jaksik, R.; Iwanaszko, M.; Rzeszowska-Wolny, J.; Kimmel, M. Microarray experiments and factors which affect their reliability. Direct. 2015, 10, 46.
- K562 cells were differentiated with PMA. However, did the cells truly differentiate into megakaryocyte lineage? The authors are requested to provide ploidy analysis and CD41 and GP1b expression levels the differentiated cells used for microarray analysis
>> We appreciate this valuable comment. K562 cells were differentiated into megakaryocytes by PMA treatment, which is widely used for megakaryocyte commitment and differentiation [Huang et al., 2014, Bütler et al., 1990]. For the present study, we differentiated K562 cells into megakaryocytes using 1 nM PMA for 7 days. To verify true differentiation into megakaryocytes, we conducted qPCR to observe the expression of megakaryocyte markers, such as GATA1, GATA2, CD34, and c-KIT, and cytological examination to determine differentiation into mature megakaryocytes. These results were indicated in the Results section (Lines 69-71 and 77-78) and Figure 1. Additionally, we analyzed megakaryocyte polyploidization, which is a main feature of megakaryocytes, as per the reviewer’s comment. Megakaryocyte ploidy analysis was completed using propidium iodide staining and measured by FACS, which method was described in the Materials and Methods section (Lines 365-367). PMA-induced megakaryocytes showed increased polyploidization level and this result was presented in the Results section (Lines 71-72) and Figure 1B. As per the reviewer’s comment, we evaluated CD41 and GP1B expression levels; however, these genes were not upregulated between K562 and megakaryocytes, as observed in microarray datasets. These results were also presented in the Results section (Lines 78-80) and Figure 1C. In this study, various major characteristics of megakaryocytes including marker gene expression, cell size, ploidy, and adherence were verified in PMA-induced megakaryocytes of K562 cells, finally indicating that K562 cells were truly differentiated into megakaryocytes after 1 nM PMA treatment for 7 days.
- Huang, R.; Zhao, L.; Chen, H.; Yin, R.H.; Li, C.Y.; Zhan, Y.Q.; Zhang, J.H.; Ge, C.H.; Yu, M.; Yang, X.M. Megakaryocytic differentiation of K562 cells induced by PMA reduced the activity of respiratory chain complex IV. PLoS One 2014, 9, e96246
- Bütler, T.M.; Ziemiecki, A.; Friis, R.R. Megakaryocytic differentiation of K562 cells is associated with changes in the cytoskeletal organization and the pattern of chromatographically distinct forms of phosphotyrosyl-specific protein phosphatases. Cancer Res. 1990, 50, 6323-6329.
- The authors need to confirm roles of important genes which were identified in the microarray analysis in the differentiation process through knock downs
>> We appreciate the reviewer’s suggestion. This study aims to understand the multifunctionality of megakaryocytes through microarray analysis in terms of bioinformatics. It will be very interesting to examine the roles of essential genes identified in the microarray analysis through knockdown experiments, as proposed by the reviewer. Therefore, further studies will be needed to understand their roles in megakaryocytes. We have discussed the implications of this further study in the Discussion section (Lines 343-345).
- CD41 and GP1b are very import megakaryocyte markers. It is surprising that these genes are not up-regulated in the datasets. The authors need to justify this phenomenon
>> We appreciate the reviewer’s suggestion. As per the reviewer’s comment, CD41 and GP1B are important markers for late-stage differentiation of megakaryocytes, including polyploidization and platelet production [Bluteau et al., 2009]. However, they were not upregulated in the microarray datasets and qPCR results, which we performed in this study. This phenomenon can be considered the following possibility: CD41 and GP1B expression was promoted by a transcription factor GATA [Ludlow et al., 1996; Jackers et al., 2004]. Although CD41 and GP1B mRNA expression was not upregulated at the time point which we observed, GATA1 and GATA2 expression was significantly increased in megakaryocytes, accompanied by enlarged cell size and multilobed nucleus. These results were indicated in Figure 1, showing the K562 differentiation into mature megakaryocytes. Particularly, it has been reported that CD41, GP1B, and ITGB3 are needed for glycoprotein signaling for platelet production rather than megakaryocyte maturation [Sun et al., 2019; Lentaigne et al., 2016]. Additionally, there are some reports that CD41 and GP1B expression was detected in K562 cells that have not differentiated into megakaryocytes or somewhat controversial in PMA-differentiated K562 cells [Rouillard et al., 2016; Conde et al., 2016]. In this study, the expression of these genes was reduced or remained unchanged in the microarray datasets and qPCR results. However, all results which we conducted to verify megakaryocyte differentiation showed that PMA-induced megakaryocytes used for this study were well differentiated from K562 cells into mature megakaryocytes as late-stage megakaryocytes with granule formation, but not the most finally mature cells for platelet production. We have described these possible points in the Discussion section (Lines 273-285) with References below.
- Bluteau, D.; Lordier, L.; Di Stefano, A.; Chang, Y.; Raslova, H.; Debili, N.; Vainchenker, W. Regulation of megakaryocyte maturation and platelet formation. Thromb. Haemost. 2009, Suppl 1, 227-234.
- Ludlow, L.B.; Schick, B.P.; Budarf, M.L.; Driscoll, D.A.; Zackai, E.H.; Cohen, A.; Konkle, B.A. Identification of a mutation in a GATA bindingsite of the platelet glycoprotein Ibbeta promoter resulting in the Bernard-Soulier syndrome. Biol. Chem. 1996, 271, 22076-22080.
- Jackers, P.; Szalai, G.; Moussa, O.; Watson, D.K. Ets-dependent regulation of target gene expression during megakaryopoiesis. Biol. Chem. 2004, 279, 52183-52190.
- Sun, S.; Jin, C.; Li, Y.; Si, J.; Cui, Y.; Rondina, M.T.; Tang, F.; Wang, Q.F. Transcriptional and spatial heterogeneity of mouse megakaryocytes at single-cell resolution. Blood2019, 134, 275.
- Lentaigne, C.;Freson, K.; Laffan, M.A.; Turro, E.; Ouwehand, W.H.; BRIDGE-BPD Consortium and the ThromboGenomics Consortium. Inherited platelet disorders: toward DNA-based diagnosis. Blood 2016, 127, 2814-2823.
- Rouillard, A.D.; Gundersen, G.W.; Fernandez, N.F.; Wang, Z.; Monteiro, C.D.; McDermott, M.G.; Ma'ayan, A. The harmonizome: a collection of processed datasets gathered to serve and mine knowledge about genes and proteins. Database (Oxford). 2016, 2016, baw100.
- Conde, I.; Pabón, D.; Jayo, A.; Lastres, P.; González-Manchón, C. Involvement of ERK1/2, p38 and PI3K in megakaryocytic differentiation of K562 cells. J. Haematol. 2010, 84, 430-440.

Reviewer 3 Report
All blood cells are derived from hematopoietic stem cells in the bone marrow. Due to a limited life time, they are continuously replenished and thus need to differentiate from the stem cell to the mature cell, ready to enter the blood stream. Megakaryocytes, the immediate precursor cells of blood platelets are extremely rare cells and their differentiation is yet poorly understood. The study entitled "Bioinformatics of differentially expressed genes in megakaryocytes by microarray analysis" by Lee, Park and Kim aims to provide insight into this process by differentiating the K562 cell line into the megakaryocytic lineage.
Major comments
While the topic remains of interest, this study is of limited interest. I have two major points of concern to start with: (1) The choice of K562 cells as a single source for their study is problematic: This cell line is derived from a blast crisis from a Bcr-Abl-positive chronic leukemia and sometimes considered to have mostly M6/erythroleukemic features. This cell line is used in the field as TPA can trigger some megakaryocytic features, but is far from "megakaryocytic" as outlined in the title. Moreover, this cell line fails to show upregulation of CD41 in their approach, which should have alerted the authors. Downregulation of CD235a (glycophorin A), cell size, adherence etc. has not been investigated Any other cell line (Meg01, M07e, CHRF) is lacking as an independent control (2) The authors claim that single cell sequencing approaches are state of the art, but too laboursome. Instead, the authors use not even bulk RNA seq approaches, but Affymetrix chips. While currently MK RNA signatures of primary human MKs are compared to those in disease conditions (sepsis, Covid-19) are published and expression levels deposited, this study generates a valid, but overall meaningless dataset.
A final point might be added: The authors could have used other described triggers like DMSO to again check for glycophorin A expression, cell size, adherence or other features to ensure that their starting position is valid.
Author Response
Reviewer #3
All blood cells are derived from hematopoietic stem cells in the bone marrow. Due to a limited life time, they are continuously replenished and thus need to differentiate from the stem cell to the mature cell, ready to enter the blood stream. Megakaryocytes, the immediate precursor cells of blood platelets are extremely rare cells and their differentiation is yet poorly understood. The study entitled "Bioinformatics of differentially expressed genes in megakaryocytes by microarray analysis" by Lee, Park and Kim aims to provide insight into this process by differentiating the K562 cell line into the megakaryocytic lineage.
Major comments
While the topic remains of interest, this study is of limited interest. I have two major points of concern to start with:
(1) The choice of K562 cells as a single source for their study is problematic: This cell line is derived from a blast crisis from a Bcr-Abl-positive chronic leukemia and sometimes considered to have mostly M6/erythroleukemic features. This cell line is used in the field as TPA can trigger some megakaryocytic features, but is far from "megakaryocytic" as outlined in the title. Moreover, this cell line fails to show upregulation of CD41 in their approach, which should have alerted the authors. Downregulation of CD235a (glycophorin A), cell size, adherence etc. has not been investigated Any other cell line (Meg01, M07e, CHRF) is lacking as an independent control
>> We appreciate this valuable comment.
As suggested by the reviewer, Meg01 and M07e cell lines are good cells used to the features of megakaryocytes because they were established from patients with chronic myelogenous and acute megakaryoblastic leukemia, respectively. Notably, K562 has also been reported to be a human leukemic cell line used as a model of hematopoietic differentiation. It can be differentiated into different hematopoietic lineage cells by various differentiation-inducing agents, such as sodium butyrate, hemin, retinoic acid, DMSO, and PMA [Sutherland et al., 1986]. Among these agents, the potent agent PMA has been used for K562 cells to differentiate into megakaryocytic cell lineage, showing that K562 cell line is a common progenitor model for megakaryocytes [Huo et al., 2006; Kim et al., 2001]. Based on previous reports, therefore, we used K562 cells to differentiate megakaryocytes using PMA. We have described this in the Discussion section (Lines 264-269) with References below. Unfortunately, we have not prepared Meg01 or M07e cell lines, as independent controls.
- Sutherland, J.A.; Turner, A.R.;Mannoni, P.; McGann, L.E.; Turc, J.M. Differentiation of K562 leukemia cells along erythroid, macrophage, and megakaryocyte lineages. Biol. Response Mod. 1986, 5, 250-262.
- Huo, X.F.; Yu, J.; Peng, H.; Du, Z.W.; Liu, X.L.; Ma, Y.N.; Zhang, X.; Zhang, Y.; Zhao, H.L.; Zhang, J.W. Differential expression changes in K562 cells during the hemin-induced erythroid differentiation and the phorbol myristate acetate (PMA)-induced megakaryocytic differentiation. Cell. Biochem. 2006, 292, 155-167.
- Kim, K.W.; Kim, S.H.; Lee, E.Y.; Kim, N.D.; Kang, H.S.; Kim, H.D.; Chung, B.S.; Kang, C.D. Extracellular signal-regulated kinase/90-KDA ribosomal S6 kinase/nuclear factor-kappa B pathway mediates phorbol 12-myristate 13-acetate-induced megakaryocytic differentiation of K562 cells. Biol. Chem. 2001, 276, 13186-13191.
Megakaryocytes are characterized by the expression of different megakaryocyte surface markers and specific transcription factor genes, and the change of DNA ploidy, aggregation, cell size, and adhesion capacity [Noetzli et al., 2019]. Among these features, we noticed that GATA1 and GATA2 expression was significantly enhanced, whereas CD34 and c-KIT expression was reduced in PMA-induced megakaryocytes. Additionally, we found to have an increased size and multilobed nuclei in PMA-induced megakaryocytes. These results are indicated in Figure 1. We also evaluated CD41 expression level; however, CD41 was not upregulated between K562 and megakaryocytes, as observed in microarray datasets. This result was presented in Figure 1 as well. Although CD41 is a marker for late-stage differentiation of megakaryocytes, including polyploidization and platelet production [Bluteau et al., 2009], it has been reported that CD41 is needed for glycoprotein signaling for platelet production rather than megakaryocyte maturation [Sun et al., 2019; Lentaigne et al., 2016]. Additionally, there are some reports that CD41 expression was detected in K562 cells that have not differentiated into megakaryocytes or somewhat controversial in PMA-differentiated K562 cells [Rouillard et al., 2016; Conde et al., 2016]. On the other hand, CD41 expression was promoted by a transcription factor GATA [Jackers et al., 2004]. Although CD41 mRNA expression was not upregulated at the time point which we observed, GATA1 and GATA2 expression was significantly increased in megakaryocytes, accompanied by enlarged cell size and multilobed nucleus. These results indicated that PMA-induced megakaryocytes used for this study were well differentiated from K562 cells into mature megakaryocytes as late-stage megakaryocytes with granule formation, not the most finally mature cells for platelet production. We have presented these results in Figure 1 and described these points in the Discussion section (Lines 269-285 and 288-290) with References below. Additionally, as the reviewer’s comment, we analyzed the expression of CD235a, which is expressed in erythroid precursors and erythrocytes, and observed the unchanged CD235a expression in PMA-induced megakaryocytes compared to K562 cells. This result was added in the Results section (Lines 81-83) and Supplementary Figure S1.
- Noetzli, L.J.; French, S.L.; Machlus, K.R. New insights into the differentiation of megakaryocytes from hematopoietic progenitors. Thromb. Vasc. Biol. 2019, 39, 1288-1300.
- Bluteau, D.; Lordier, L.; Di Stefano, A.; Chang, Y.; Raslova, H.; Debili, N.; Vainchenker, W. Regulation of megakaryocyte maturation and platelet formation. Thromb. Haemost. 2009, Suppl 1, 227-234.
- Sun, S.; Jin, C.; Li, Y.; Si, J.; Cui, Y.; Rondina, M.T.; Tang, F.; Wang, Q.F. Transcriptional and spatial heterogeneity of mouse megakaryocytes at single-cell resolution. Blood2019, 134, 275.
- Lentaigne, C.;Freson, K.; Laffan, M.A.; Turro, E.; Ouwehand, W.H.; BRIDGE-BPD Consortium and the ThromboGenomics Consortium. Inherited platelet disorders: toward DNA-based diagnosis. Blood 2016, 127, 2814-2823.
- Rouillard, A.D.; Gundersen, G.W.; Fernandez, N.F.; Wang, Z.; Monteiro, C.D.; McDermott, M.G.; Ma'ayan, A. The harmonizome: a collection of processed datasets gathered to serve and mine knowledge about genes and proteins. Database (Oxford). 2016, 2016, baw100.
- Conde, I.; Pabón, D.; Jayo, A.; Lastres, P.; González-Manchón, C. Involvement of ERK1/2, p38 and PI3K in megakaryocytic differentiation of K562 cells. J. Haematol. 2010, 84, 430-440.
- Jackers, P.; Szalai, G.; Moussa, O.; Watson, D.K. Ets-dependent regulation of target gene expression during megakaryopoiesis. Biol. Chem. 2004, 279, 52183-52190.
(2) The authors claim that single cell sequencing approaches are state of the art, but too laboursome. Instead, the authors use not even bulk RNA seq approaches, but Affymetrix chips. While currently MK RNA signatures of primary human MKs are compared to those in disease conditions (sepsis, Covid-19) are published and expression levels deposited, this study generates a valid, but overall meaningless dataset.
>> Thank you for this insightful question. Based on the reviewer’s comment, RNA-seq has the advantage of identifying single-nucleotide polymorphisms because it can perform a sequence-based analysis. Additionally, RNA-seq is more sensitive and accurate than microarrays. However, there are many limitations, such as large amounts of data, the complexity of analysis, non-optimized protocol, and almost importantly, high cost. They should be overcome to become the predominantly used method for transcriptome analysis. Alternatively, microarrays are still used in clinical diagnostic testing for diseases because they are a well-established process and cost-effective for large-scale studies. Therefore, we choose microarrays rather than RNA-seq, as a more universal and accessible technology, to demonstrate the general possibility of megakaryocyte multifunctionality in terms of bioinformatics. In the Introduction section (Lines 52-60), we have described this point with References below.
- Rodriguez-Esteban, R.; Jiang, X. Differential gene expression in disease: a comparison between high-throughput studies and the literature. BMC Med. Genomics 2017, 10, 59.
- Zhao, S.; Fung-Leung, W.P.; Bittner, A.; Ngo, K.; Liu, X. Comparison of RNA-Seq and microarray in transcriptome profiling of activated T cells. PLoS ONE 2014, 9, e78644.
- Lemuth, K.; Rupp, S. Microarrays as research tools and diagnostic devices. RNA and DNA Diagnostics, Springer: Switzerland, 2015; 259-280.
- Jaksik, R.; Iwanaszko, M.; Rzeszowska-Wolny, J.; Kimmel, M. Microarray experiments and factors which affect their reliability. Direct. 2015, 10, 46.
Moreover, we agree that most of the microarray results we provided indicate no significant changes or are not easy to interpret. We have mentioned genes that are only a few dozen of the more than 40,000 genes. However, these genes were included in clusters related to megakaryocyte function, indicating significant expression levels. Even if many meaningless genes are excluded, this study suggests that the functions of megakaryocytes can be suggested from the changes in the genes we have adopted.
A final point might be added: The authors could have used other described triggers like DMSO to again check for glycophorin A expression, cell size, adherence or other features to ensure that their starting position is valid.
>> We appreciate the reviewer’s comment. K562 is a human leukemic cell line used as a model of hematopoietic differentiation and there are many differentiation-inducing agents to differentiate K562 into megakaryocytes [Sutherland et al., 1986]. Among them, PMA-induced differentiation of K562 cells was a classic model to study the megakaryocytic differentiation of blood cells. This process is followed by the alterations in cell morphology, the acquisition of adhesion properties, cell growth arrest, and specific markers expressed on the cell surface of megakaryocytes [Huang et al., 2014; Bütler et al., 1990]. Based on these reports, we selected PMA-induced megakaryocytes of K562 cells for this study. We have described this point in the Discussion section (Lines 264-269) with References below. Among various features for megakaryocytes, we noticed that GATA1 and GATA2 expression was significantly enhanced, whereas CD34 and c-KIT expression was reduced in PMA-induced megakaryocytes. Additionally, we found to have an increased size and multilobed nuclei with polyploidization in PMA-induced megakaryocytes. As per the reviewer’s comment, we compared cell adhesion because megakaryocytic differentiation is accompanied by increased cell adhesion properties. PMA-induced megakaryocytes had increased adherence to the culture plates and were more spread out compared with K562 cells. Finally, these results indicated that K562 cells were truly differentiated into megakaryocytes, as presented in the Results section (Lines 69-73 and 77-80), Figure 1, and Supplementary Figure S1.
- Sutherland, J.A.; Turner, A.R.;Mannoni, P.; McGann, L.E.; Turc, J.M. Differentiation of K562 leukemia cells along erythroid, macrophage, and megakaryocyte lineages. Biol. Response Mod. 1986, 5, 250-262.
- Huang, R.; Zhao, L.; Chen, H.; Yin, R.H.; Li, C.Y.; Zhan, Y.Q.; Zhang, J.H.; Ge, C.H.; Yu, M.; Yang, X.M. Megakaryocytic differentiation of K562 cells induced by PMA reduced the activity of respiratory chain complex IV. PLoS One 2014, 9, e96246
- Bütler, T.M.; Ziemiecki, A.; Friis, R.R. Megakaryocytic differentiation of K562 cells is associated with changes in the cytoskeletal organization and the pattern of chromatographically distinct forms of phosphotyrosyl-specific protein phosphatases. Cancer Res. 1990, 50, 6323-6329.

Round 2
Reviewer 3 Report
realizing the point-to-point letter that the authors have provided, I have to state that I will still consider this manuscript as to be rejected, as the authors were not willing to provide new experiments or new analyses of previous data. The revised manuscript is not suitable for publication, but I kindly ask to remove my previous (doubled) letter.